# Ultrasonographic Assessment of the Diaphragm

**DOI:** 10.3390/diagnostics14141481

**Published:** 2024-07-10

**Authors:** Taiga Itagaki, Yusuke Akimoto, Takuya Takashima, Jun Oto

**Affiliations:** 1Department of Emergency and Disaster Medicine, Tokushima University Hospital, 2-50-1 Kuramoto, Tokushima 770-8503, Japan; 2Emergency Department, Tokushima Prefectural Miyoshi Hospital, 815-2 Ikedacho Shima, Miyoshi 778-0005, Japan; akkiii807@gmail.com; 3Department of Emergency and Critical Care Medicine, Tokushima University Graduate Hospital of Biomedical Sciences, 3-18-15 Kuramoto, Tokushima 770-8503, Japan; takashima.takuya.2@tokushima-u.ac.jp (T.T.); joto@tokushima-u.ac.jp (J.O.)

**Keywords:** diaphragm, inspiratory muscles, diagnostic ultrasonography, diaphragm-protective ventilation

## Abstract

Mechanical ventilation injures not only the lungs but also the diaphragm, resulting in dysfunction associated with poor outcomes. Diaphragm ultrasonography is a noninvasive, cost-effective, and reproducible diagnostic method used to monitor the condition and function of the diaphragm. With advances in ultrasound technology and the expansion of its clinical applications, diaphragm ultrasonography has become increasingly important as a tool to visualize and quantify diaphragmatic morphology and function across multiple medical specialties, including pulmonology, critical care, and rehabilitation medicine. This comprehensive review aims to provide an in-depth analysis of the role and limitations of ultrasonography in assessing the diaphragm, especially among critically ill patients. Furthermore, we discuss a recently published expert consensus and provide a perspective for the future.

## 1. Introduction

Although it is known that mechanical ventilation causes lung injury, it is less widely appreciated that ventilation also injures the diaphragm as well [1,2,3]. Controlled ventilation for patients with acute respiratory failure leads to muscle atrophy in the diaphragm due to oxidative stress-mediated protein degradation [4,5]. Since Levine et al. [6] reported in 2008 that short-duration controlled ventilation in brain-dead organ donors causes marked atrophy of diaphragm myofibers, more studies using ultrasonography have shown that diaphragm atrophy and dysfunction are frequent complications that occur during mechanical ventilation and are associated with poor outcomes [7,8,9,10]. This has led to the need for noninvasive and reproducible methods of monitoring the diaphragm’s condition and function [11].

Several technologies are currently available for this purpose; however, many have limitations, such as being invasive, less cost-effective, and difficult to use at the bedside. Diagnostic ultrasonographic devices are ideal in this regard, and they are becoming an important tool for visualizing and quantifying diaphragm morphology and function. Additionally, respiratory muscles like the diaphragm and abdominal muscles are relatively close to the body surface, making them easier to observe with ultrasonography.

This article first summarizes the mechanisms of diaphragm muscle injury and describes the diaphragm evaluation method (diaphragm ultrasonography) using ultrasonographic equipment, its clinical applications, and its limitations. Furthermore, we discuss a recently published expert consensus [12] and provide a perspective for the future.

## 2. Mechanisms of Diaphragm Muscle Injury Induced by Mechanical Ventilation

To prevent diaphragm dysfunction, it is essential to understand the mechanisms of diaphragm muscle injury. Atrophy due to a suppression of inspiratory effort and injury owing to an excessive load are the two most important diaphragm injuries during mechanical ventilation.

### 2.1. Disuse Atrophy

Disuse atrophy due to a suppression of inspiratory effort and excessive respiratory support is the most important mechanism of diaphragm injury during mechanical ventilation [13]. In animal studies, controlled ventilation or high levels of pressure support ventilation caused acute muscle atrophy, damage of myofibers, and dysfunction [4,14,15]. Levine et al. [6] reported that diaphragm inactivity for up to 18–69 h in brain-dead patients was associated with noticeable atrophy of the diaphragm; however, it did not cause atrophy in the pectoralis major. Furthermore, histological studies in humans have revealed that disuse of the diaphragm activated the proteolytic pathways, leading to both diaphragmatic atrophy and mitochondrial dysfunction, resulting in reduced contractility [16,17].

### 2.2. Concentric Load-Induced Injury

Insufficient ventilatory support against inspiratory effort overloads the diaphragm and causes muscle injury. In histological investigations in healthy subjects and patients with chronic obstructive pulmonary disease (COPD), contraction of the diaphragm against excessive load caused acute diaphragm injury, inflammation, and weakness [18,19]. Importantly, in critically ill patients with systemic inflammation, mechanical stimuli can exacerbate sarcolemma and thus contribute to diaphragmatic dysfunction [20].

### 2.3. Eccentric Load-Induced Injury

It is known that muscle injury occurs when muscles eccentrically contract during lengthening [21], and this also applies to the diaphragm [22]. This muscle injury is supposed to occur in mechanically ventilated patients. One possible cause is patient–ventilator asynchrony. With asynchronies, such as premature cycling, ineffective effort, and reverse triggering, the diaphragm will be forced to contract during the expiratory phase of the machine cycle, which results in diaphragm injury [2,23]. Another cause is post-inspiratory diaphragm activity, or so-called “diaphragm braking”. The diaphragm contracts even during the expiratory phase and suppresses the rate of decrease in lung volume to prevent acute alveolar collapse and following atelectasis [24]. This physiological contraction during the expiratory phase is potentially injurious to the diaphragm [25]. Diaphragm braking is strong in situations where alveoli are predisposed to collapse such as when the positive end-expiratory pressure (PEEP) is set low [24]. In addition, it has been suggested that the diaphragm may contract during expiration if the expiratory muscles are recruited, especially in patients with a small airway obstruction [26,27]. However, the effects of eccentric contraction that occurs with expiratory muscle recruitment on diaphragmatic function remains unknown.

### 2.4. Longitudinal Atrophy

It is hypothesized in animal studies that if the diaphragm is maintained in a contracted state at a higher PEEP, the abrupt lowering of PEEP, such as during spontaneous breathing trials, can cause longitudinal atrophy [28]. A higher PEEP shortens the length of sarcomeres (the basic contractile unit of muscle fibers composed of two main protein filaments, actin and myosin) in the longitudinal direction, and gradually, some sarcomeres drop out and others regain their original length (reconstruction). Here, the sudden lowering of the PEEP causes overstretching of the sarcomeres, leading to a change in the length–tension relationship of the diaphragm that may cause impaired contractility.

## 3. Basics of Diaphragm Ultrasonography

There are two approaches to diaphragm ultrasonography. One is the intercostal approach at the zone of apposition (the origin of the diaphragm in contact with the inner surface of the rib cage), and the other is the subcostal approach with the liver as the acoustic window. The former is primarily used to assess diaphragm thickness and contraction, while the latter is used to assess diaphragmatic exercise. Diaphragmatic exercise can be assessed under various conditions, such as during quiet breathing, maximum inspiration, and short diaphragm contractions during sniffing. Reference values for each parameter used in diaphragm ultrasonography [3,29,30,31,32,33,34,35,36,37] are shown in Table 1. For intensive care unit (ICU) patients, cutoff values that showed differences in clinical outcomes, such as success in spontaneous breathing trials (SBTs) or weaning from mechanical ventilation, are indicated.

### 3.1. Intercostal Approach

The intercostal approach involves positioning a 10–15 MHz linear transducer vertically along the cephalocaudal axis at the right 8th–11th intercostal space on the anterior or mid-axillary line in the zone of apposition [38,39]. The diaphragm appears at a depth of 2–4 cm as a layered structure between the pleura and peritoneum, with a white linear structure in the center (Figure 1A,B). During measurements, the thickness of the pleura and peritoneum should not be included. The normal diaphragm thickness in healthy individuals is approximately 1.6 mm (1.9 mm in males and 1.4 mm in females) [18].

The diaphragm thickens upon contraction, and the thickening fraction of the diaphragm (TFdi) is the measure of diaphragmatic contraction activity [40]. The TFdi can be determined using B-mode (Figure 1A) or M-mode (Figure 1B) as the percentage increase in diaphragm thickness during inspiration: (end-inspiratory diaphragm thickness−end-expiratory diaphragm thickness)/end-expiratory diaphragm thickness × 100. The TFdi is around 37% in healthy individuals at rest [20], but it shows considerable variability during maximum breathing efforts [17,24,41].

Measuring diaphragm thickness is subject to technical and methodological limitations. The diaphragm is extremely thin, around 1.5–2.0 mm, so slight measurement errors can lead to an overestimation or underestimation of diaphragm thickness and the TFdi. Goligher et al. [40] reported that marking the transducer position on the skin reduced the absolute difference in end-expiratory diaphragm thickness to 0.2 mm and lower within the same examiner and to 0.4 mm and lower between different examiners. Notably, this difference corresponds to about 10% of the diaphragm thickness. Additionally, it is important to note that the TFdi does not account for the duration of diaphragm contraction (inspiratory time) or contraction frequency (respiratory rate), nor does it consider the involvement of accessory inspiratory and expiratory muscles [35].

### 3.2. Subcostal Approach

To observe diaphragmatic movement using the subcostal approach, a low-frequency (2–5 MHz) transducer is placed just below the costal margin along the midclavicular line. The patient should be in a semi-recumbent position, and the ultrasound beam should be directed as cranially as possible, perpendicular to the diaphragm dome (Figure 2). In this view, the diaphragm appears as a bright line covering the liver or spleen. The right side is easier to image, since the liver is an acoustic window. Conversely, the spleen does not serve as an effective acoustic window, making it challenging to obtain clear images of the left diaphragm. This approach allows for the visualization of diaphragmatic exercise in over 95% of cases during quiet breathing but becomes difficult during maximum respiration, especially on the left side [35]. Da Conceicao D et al. recently explored a new approach to assess diaphragmatic motion by measuring the excursion of the uppermost point of the zone of apposition at the mid-axillary line using a high-frequency linear transducer and reported that this new approach had a higher success rate bilaterally (both 100%) than the subcostal approach (98.7% on the right side and 34.7% on the left side) [42].

Typically, the diaphragm moves toward the transducer during inspiration. To quantify this exercise, place the M-mode line perpendicular to the direction of the exercise and measure the excursion distance. Setting the sweep speed to about 10 mm/second allows for the visualization of approximately three respiratory cycles within a single image.

However, excursion measurement is only feasible during unassisted spontaneous breathing (e.g., with a T-piece or low continuous positive airway pressure [CPAP]). This limitation arises because it is impossible to distinguish between diaphragm exercise due to spontaneous contraction and that due to inspiratory pressure support from a mechanical ventilator, and because excursion is significantly influenced by the lung volume [38,43]. Therefore, excursion cannot be used to evaluate the work of breathing, and the TFdi can work for this purpose.

## 4. Clinical Applications of Diaphragm Ultrasonography

### 4.1. Evaluation of Diaphragm Thickness

Diaphragm atrophy occurs early after the initiation of mechanical ventilation and is associated with diaphragm dysfunction [4,5,6]. Zambon et al. [44] demonstrated a linear correlation between the duration of mechanical ventilation and the rate of diaphragm atrophy, with a decrease of 7.5% per day under controlled ventilation and an increase of 2.3% per day during spontaneous breathing or CPAP. Previous studies [7,45] reported that the end-expiratory diaphragm thickness decreased by >10% in 41% and 63% of patients and increased by >10% in 24% and 19% of patients, with both groups experiencing prolonged mechanical ventilation and higher mortality rates. In both studies, a decrease in the thickness of the diaphragm was observed until the third day of mechanical ventilation and during both controlled and partially assisted ventilation. These findings suggest the importance of monitoring changes in diaphragm thickness over time.

### 4.2. Evaluation of Diaphragm Function

Diaphragm dysfunction is a common issue that can impact the outcomes of ICU patients [8,9,10]. It is often indicated by an excursion of <10 mm during resting breathing or a TFdi of <20% during maximal breathing [34,35]. In cases of unilateral diaphragmatic paralysis, both the excursion and TFdi of the paralyzed side decrease [36], making diaphragm ultrasonography a suitable diagnostic tool. However, it is crucial to observe both sides, including the left diaphragm, which can be challenging to visualize.

In emergency settings, noninvasive ventilation (NIV) is used for patients with acute exacerbations of COPD. An improvement in excursion 1 h after initiating NIV has been associated with successful NIV outcomes [46]. Lung hyperinflation due to air trapping can restrict diaphragmatic movement [47]; hence, evaluating diaphragm function can help assess NIV effectiveness and potentially prevent delays in intubation for patients with acute exacerbations of COPD.

### 4.3. Evaluation of Respiratory Effort

The proper monitoring of respiratory effort in mechanically ventilated patients is crucial for optimizing diaphragm activity and ensuring lung and diaphragm protection [25]. Although specific indicators for diaphragm-protective respiratory efforts are not yet established, the levels observed in healthy individuals and patients who have been successfully weaned from mechanical ventilation serve as a reference value (Table 1) [48]. The TFdi correlates with invasive measures like transdiaphragmatic pressure (Pdi) [40,49,50] and the pressure–time product (PTP) of esophageal pressure, making it useful for noninvasive monitoring of the work of breathing.

In contrast, Oppersma et al. [51], in a study examining the diaphragm evaluation capability of speckle tracking (see below), demonstrated that the Pdi and diaphragm muscle potentials did not correlate with the TFdi when 13 healthy volunteers were given a respiratory load of up to 50% of the maximum inspiratory pressure. Recent studies in mechanically ventilated patients [52] also showed a weak correlation between the TFdi and Pdi (ρ = 0.11, *p* = 0.008) and no correlation with the transdiaphragmatic PTP (ρ = 0.04, *p* = 0.396). Thus, the TFdi has limitations as a measure of the work of breathing. These discrepancies arise because the TFdi does not account for the time components (inspiratory time and respiratory rate) that are integral to PTP, and while the TFdi directly reflects diaphragm activity, the Pdi is influenced by factors other than diaphragmatic activity [53].

### 4.4. Prediction of Weaning Outcomes

There have been many attempts to predict weaning outcome by performing diaphragm ultrasonography during SBT. DiNino et al. [29] reported that a ΔTFdi ≥ 30% during SBT had a sensitivity of 88% and a specificity of 71% (area under the curve [AUC] 0.79) for predicting extubation failure (reintubation or NIV use within 48 h), which surpassed the predictive ability of the rapid shallow breathing index (RSBI). On the other hand, it has been reported that an excursion < 10 mm before the start of SBT did not predict extubation failure (AUC 0.61) [3], whereas an excursion < 10 mm 30 min after the start of SBT predicted extubation failure at a high rate [31].

There have been attempts to enhance the predictive ability by considering the time component of excursion. Spadaro et al. [54] defined diaphragmatic RSBI (D-RSBI) as the respiratory rate divided by the excursion (mm), reporting that a D-RSBI > 1.3 during SBT predicted weaning failure (SBT failure or reintubation/NIV use within 48 h) with a high probability compared to RSBI alone (AUC: D-RSBI 0.89, RSBI 0.72). Palkar et al. [55] introduced the excursion–time (E-T) index, calculated by multiplying the excursion (cm) by the inspiratory time (seconds), and reported that a difference of <3.8% between the E-T index during assisted control ventilation and after the start of SBT predicted extubation failure with a sensitivity of 79.2% and a specificity of 75%.

Indeed, the frequency of diaphragm dysfunction during the weaning phase is high, ranging from 30% to 70% [3,8,43,56], but many cases are successfully extubated [57,58,59]. The excursion of 191 cases who successfully underwent SBT with a T-piece showed no correlation with reintubation within 7 days following extubation, with only the cough intensity being related to successful extubation [56]. While noting the variation in the definition of extubation failure across studies and the differing management of patients who failed SBT or those who were extubated in a palliative manner, the authors concluded that the success of extubation after successful SBT is determined by factors other than diaphragm function.

Additionally, Mayo et al. [60] argued that although diagnostic ultrasonography is a useful bedside tool in critical care settings, identifying the factors contributing to weaning failure through a single ultrasonographic examination is difficult, emphasizing the importance of cross-sectional evaluation of the heart, lungs, diaphragm, and pleural space (including pleural effusion). Similarly, Tuinman et al. [39] proposed the ABCDE ultrasound approach, a systematic ultrasound evaluation of the heart, lungs, and respiratory muscle pump, in patients with weaning failure.

## 5. Expert Consensus

Considering the rapid dissemination of diaphragm ultrasonography in recent years, an expert consensus utilizing the Delphi method with 14 panelists (Table 2) was published in 2022 to establish high-quality protocols and recommendations [12]. Discussions were held on 75 questions across seven domains: anatomy and physiology, transducer settings, technique, the effects of mechanical ventilation, learning and expertise, daily practice, and future directions. A convergence of opinions was observed in 61% of the questions, demonstrating agreement on high-quality and homogeneous measurement methods and highlighting areas lacking consensus, namely questions requiring further research.

Initially, unresolved measurement issues encountered in daily practice, such as at what height within the zone of apposition to measure the diaphragm thickness and which mode to use for the TFdi measurement (Figure 1A,B), did not reach a unified consensus. Furthermore, cutoff values for the increase in diaphragm thickness from a histological perspective were not determined due to the difficulty [61] in distinguishing between muscle hypertrophy and inflammation, edema, and fibrosis ultrasonographically. Additionally, cutoff values for diaphragm dysfunction based on the TFdi were not specified due to the likelihood of measurement errors; moreover, there were possible variations according to measurement conditions (with or without ventilatory assistance, during respiratory distress, during SBT, etc.) or outcomes that we want to predict (SBT or extubation failure and inadequate or excessive respiratory assistance).

## 6. Future Perspectives

The expert consensus in the preceding section also summarized the prospects and research topics regarding diaphragm ultrasonography.

In recent years, the expectations for diaphragm ultrasonography using new technologies have been rising. Speckle tracking is one such example. This method involves tracking the position of speckles on ultrasonographic images to quantify tissue strain and assess motion and deformation [39]. Unlike tissue doppler imaging, speckle tracking does not require angle correction for motion direction and can measure the strain and strain rate in two directions. Oppersma et al. [51] used speckle tracking to measure the diaphragm strain and strain rate (acceleration of deformation) when healthy individuals were subjected to respiratory loads of up to 50% of the maximum inspiratory pressure. They found that both the strain and strain rate were strongly correlated with the Pdi and electromyogram of the diaphragm. Goutman et al. [62] also reported that speckle tracking could accurately measure excursion in two directions, unlike M-mode, and quantify the movement of the left diaphragm.

Shear wave elastography is a technique for quantifying tissue elasticity. As changes in muscle elasticity due to injuries or fibrosis may reflect functional impairment, its application to the diaphragm is being investigated. A study involving 15 healthy individuals showed that changes in diaphragm elasticity evaluated through shear wave elastography were correlated with changes in the Pdi [63].

## 7. Conclusions

Diaphragm ultrasonography is a minimally invasive, versatile examination that can be performed at the bedside. Diaphragm ultrasonography enables the diagnosis of conditions such as phrenic nerve palsy and diaphragm dysfunction, assessing the respiratory effort of lung- and diaphragm-protective ventilation and predicting the success or failure of weaning from mechanical ventilation. However, no clinical studies demonstrate that diaphragm ultrasonography improves patient outcomes; thus, further research is needed to determine effective utilization methods that lead to meaningful therapeutic interventions.

## Figures and Tables

**Figure 1 diagnostics-14-01481-f001:**
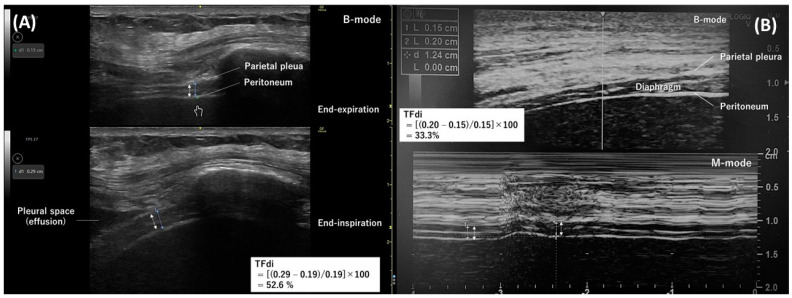
Diaphragm ultrasonography with the intercostal approach. The double arrow indicates the thickness of the diaphragm. (**A**) A B-mode diaphragm ultrasonographic image obtained with a linear transducer along the right mid-axillary line. Separate images measuring the diaphragm thickness at end-expiration (**top**) and end-inspiration (**bottom**) were used to determine the thickening fraction of the diaphragm. (**B**) An M-mode ultrasonographic image (**bottom**) capturing temporal changes in one direction of the thickness of the diaphragm. The diaphragm thickness at end-expiration and end-inspiration were measured from this image to determine the thickening fraction of the diaphragm.

**Figure 2 diagnostics-14-01481-f002:**
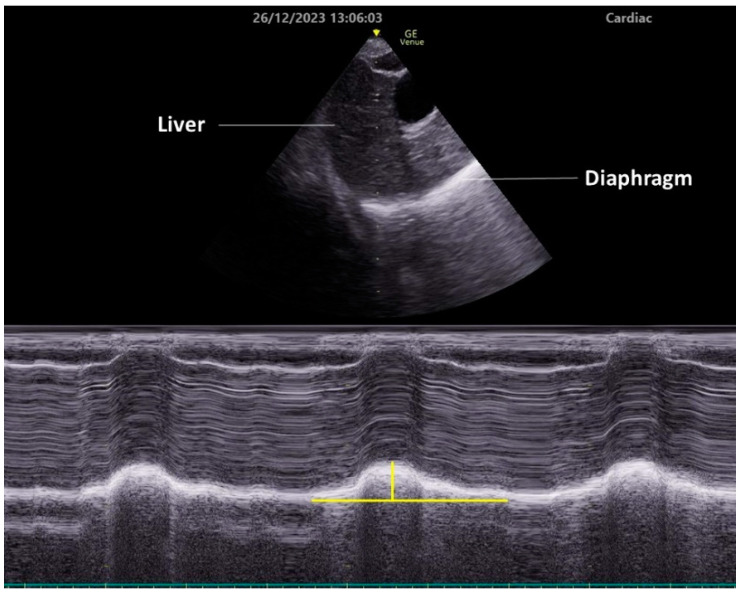
Diaphragm ultrasonography with the subcostal arch approach. An M-mode diaphragm ultrasonographic image was obtained using a cardiac transducer positioned subcostally on the right side and directed cranially through the liver. The lower image shows the diaphragm displacement passing through the M-mode beam over time. Vertical yellow line indicates diaphragm excursion.

**Table 1 diagnostics-14-01481-t001:** Parameters and reference values for diaphragm ultrasonography.

	Parameter	Reference Value	Abnormal Values ※	Cutoff Value Related to Outcome	References
Healthy individual	End-expiratory thickness (mm)	male: 1.9 ± 0.4	male: <1.7		[34]
		female: 1.4 ± 0.3	female: <1.3		
	TFdi	37 ± 9%	<20%		[36]
		80 ± 50%	<20%		[37]
		(maximal breathing)			
	Excursion (mm)	male: 18 ± 3	male: <10		[35]
		female: 16 ± 3	female: <9		
ICU patient	End-expiratory thickness (mm)			<1.7	[29]
	TFdi			<30~36%	[29,30,31]
	Excursion (mm)			<10~14 <25 (maximal breathing)	[3,32,33]

※ Lower limit of 95% confidence interval or minimum value. ICU, intensive care unit; TFdi, thickening fraction of the diaphragm.

**Table 2 diagnostics-14-01481-t002:** Expert consensus.

**Anatomy and physiology**
The significance of echogenicity is unknown but should be investigated
≥10% decrease from baseline thickness is regarded as the cutoff for clinically relevant atrophy
Obesity and large tidal volume can complicate measurements
Maximum effort measurements offer important information but are hard to obtain and compare due to the subjectivity of a maximum effort
A diaphragm excursion <2 cm is indicative of dysfunction during quiet breathing
※ No consensus was achieved The continuity of diaphragm thickness in the zone of apposition The cutoff for increased thickness due to confounding with inflammation and edema The cutoff for dysfunction based on thickening fraction
**Ventilator impact**
Positive pressure ventilation augments amplitude with greater lung inflation PEEP lowers the diaphragm resting position and reduces excursion
Positive pressure ventilation reduces patient effort and, as such, the thickness at end-inspiration PEEP lowers the diaphragm resting position with a higher thickness at end-expiration due to shortening of the muscle
Positive pressure ventilation reduces patient effort and, as such, diaphragmatic thickening PEEP lowers the diaphragm resting position with a higher thickness at end-expiration due to the shortening of the muscle and, as such, its percentual thickening
**Transducer settings and technique**
*Excursion* The ideal range is between 2 and 5 MHz (cardiac or abdominal transducer) The ideal mode is the M-mode The maximum depth should be adjusted to capture the maximum excursion The gain should be adjusted to create ideal contrast with the surrounding structure The transducer should be aimed at the dome of the diaphragm Measurements are best performed in M-mode and during quiet breathing Organ displacement is a valid alternative for excursion if the diaphragm dome is hard to visualize ※ No consensus was achieved Transducer placement on the abdomen
*Thickness* The ideal range is between 7 and 12 MHz (linear transducer) The depth should be set just below to several centimeters under the diaphragm The gain should be adjusted to create an ideal contrast with the surrounding structure The transducer should be placed on the mid-axillary line or slightly more ventral, approximately between the 8th and 11th rib, with lung slightly or just not moving into the image The transducer should be placed perpendicular to the chest wall, so that all three layers (pleura, peritoneum, and fibrous layer) are visible The caliper placement should be as close as possible to the pleural and peritoneal line without including these lines in the measurement ※ No consensus was achieved Preferring B-mode or M-mode Transducer orientation to be in line with or perpendicular to the intercostal space The optimal breathing pattern for making measurements
*Both* A unilateral measurement of the diaphragm on the right side of the patient is an acceptable proxy for the whole diaphragm, unless there is any suspicion of unilateral pathology (e.g., thoracic surgery, phrenic nerve, or spinal cord injury), in which case this needs to be excluded or measurements need to be taken on both sides
**Learning and expertise**
Measuring diaphragm excursion is an easy skill with a steep learning curve
Measuring diaphragm thickness is not an easy skill and has a slow learning curve
A teaching program to learn diaphragm ultrasonography should include the anatomy of the diaphragm, anatomical landmarks for measurement, supervised practice, and a practical skill examination A minimum of 40 (ideally bilateral) examinations, of which at least 20 should be under the (indirect) supervision of an experienced teacher, are needed for independent use in daily practice
**Daily practice**
*Skills necessary in daily practice* Excursion measurements are a necessary skill for daily practice Thickness measurements to calculate diaphragm thickening are a necessary skill for daily practice
*Useful indications* Monitoring diaphragm function and determining dysfunction Prognostication of difficult weaning, extubation outcome, and length of ICU stay Detect patient–ventilator asynchrony and titrate ventilator settings

The punctuation mark (※) represents questions on which no agreement was reached. (Modified from Reference [12]).

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
