# Peer review of "Ultrasonographic Assessment of the Diaphragm"

_diagnostics, 2024, doi:10.3390/diagnostics14141481_

Round 1

Reviewer 1 Report

Comments and Suggestions for Authors

This is a thoughtful and well-written narrative review of the current state of understanding of both ventilator-induced diaphragmatic injury and diaphragmatic ultrasound.  I commend the authors on the breadth and depth of their review.

The main thing missing from this view is any mention of a recently validated third technique for assessing diaphragmatic excursion: using a linear, high-frequency probe to measure externally on the body the distance travelled by the zone of apposition (ZOA) along the mid-axillary line from end-expiration to end-inspiration during vital capacity breathing.  Here are a few relevant references:

1)    Earliest mention of this approach in the literature

https://pubmed.ncbi.nlm.nih.gov/27796838/

2)    Use of this approach in a study of nerve blocks that impair phrenic activity

https://pubmed.ncbi.nlm.nih.gov/31283740/

3)    Validation study showing lower inter-operator variability for this approach than the 2 historical approaches mentioned in this review

https://pubmed.ncbi.nlm.nih.gov/37940349/

Notably, the ZOA for these studies is defined differently than how the Authors define it.  For the studies above, ZOA is defined as the point on the lateral chest wall where both the aerated lung and the diaphragm are visible in the same rib interspace.  The advantage of this new approach for diaphragmatic ultrasound is two-fold (1) the new approach has demonstrated much higher inter-operater reliability than the two historical approaches and (2) the new approach can reliably visualize left diaphragmatic excursion in all patients whereas the low-frequency approach sometimes cannot.

Comments on the Quality of English Language

This article would benefit from review by a native English speaker to correct minor grammatical issues.  For example:

Lines 27-28: “It is known that mechanical ventilation causes lung injury, however it also injures the diaphragm as well.”  This sentence has 2 main clauses, which is non-standard English.  A more grammatically correct phrasing would be: “Although it is known that mechanical ventilation causes lung injury, it is less widely appreciated that ventilation also injures the diaphragm as well.” 

Lines 30-33: “Since Levine et al. reported in 2008 that short duration-controlled ventilation in brain-dead organ donors causes marked atrophy of diaphragm myofibers, and more studies using ultrasonography have shown that diaphragm atrophy and dysfunction are frequent complications that occur during mechanical ventilation and are associated with poor outcomes.”  The “and” in the middle needs to be removed to make this sentence align with standard English grammar as follows: “Since Levine et al. reported in 2008 that short duration-controlled ventilation in brain-dead organ donors causes marked atrophy of diaphragm myofibers, more studies using ultrasonography have shown that diaphragm atrophy and dysfunction are frequent complications that occur during mechanical ventilation and are associated with poor outcomes.” 

Lines 66-67: “Importantly, in critically ill patients with systemic inflammation, sarcolemma is 66 vulnerable to mechanical stimuli, therefore diaphragm dysfunction through this mecha-67 nism frequently happens20.”  This sentence is oddly worded and could be rephrased as something like this: “Importantly, in critically ill patients with systemic inflammation, mechanical stimuli can exacerbate sarcolemma and thus contribute to diaphragmatic dysfunction.”

Author Response

Reviewer 1

Comments and Suggestions for Authors

This is a thoughtful and well-written narrative review of the current state of understanding of both ventilator-induced diaphragmatic injury and diaphragmatic ultrasound. I commend the authors on the breadth and depth of their review.

[Response]

Thank you for your review and valuable comments. We tried to address all concerns of you by providing a point-by-point response.

The main thing missing from this view is any mention of a recently validated third technique for assessing diaphragmatic excursion: using a linear, high-frequency probe to measure externally on the body the distance travelled by the zone of apposition (ZOA) along the mid-axillary line from end-expiration to end-inspiration during vital capacity breathing. Here are a few relevant references:

1) Earliest mention of this approach in the literature

https://pubmed.ncbi.nlm.nih.gov/27796838/

2) Use of this approach in a study of nerve blocks that impair phrenic activity

https://pubmed.ncbi.nlm.nih.gov/31283740/

3) Validation study showing lower inter-operator variability for this approach than the 2 historical approaches mentioned in this review

https://pubmed.ncbi.nlm.nih.gov/37940349/

Notably, the ZOA for these studies is defined differently than how the Authors define it.  For the studies above, ZOA is defined as the point on the lateral chest wall where both the aerated lung and the diaphragm are visible in the same rib interspace.  The advantage of this new approach for diaphragmatic ultrasound is two-fold (1) the new approach has demonstrated much higher inter-operator reliability than the two historical approaches and (2) the new approach can reliably visualize left diaphragmatic excursion in all patients whereas the low-frequency approach sometimes cannot.

[Response]

Thank you for sharing this new method. The ability to evaluate bilateral diaphragm function with high reproducibility will bring new possibilities to diaphragm ultrasound. However, this method has only just been proposed and has been validated in only one single-center study (reference 3, which you provided) and is not mentioned in the expert consensus. As you see, this review focuses primarily on diaphragm ultrasonography in mechanically ventilated critical ill patients, therefore I included this method in the context of its ability to reliably visualize left diaphragm excursion as follows.

“Da Conceicao D, et al explored a new approach to assess diaphragmatic motion by measuring the excursion of the uppermost point of the zone of apposition at the mid-axillary line using a high-frequency linear transducer, and reported that this new approach had a higher success rate bilaterally (100% both) than the subcostal approach (98.7% on the right side and 34.7% on the left side).”

Comments on the Quality of English Language

This article would benefit from review by a native English speaker to correct minor grammatical issues.  For example:

Lines 27-28: “It is known that mechanical ventilation causes lung injury, however it also injures the diaphragm as well.”  This sentence has 2 main clauses, which is non-standard English.  A more grammatically correct phrasing would be: “Although it is known that mechanical ventilation causes lung injury, it is less widely appreciated that ventilation also injures the diaphragm as well.”

[Response]

Thank you for your advice. We have changed the sentence to that you suggested.

Lines 30-33: “Since Levine et al. reported in 2008 that short duration-controlled ventilation in brain-dead organ donors causes marked atrophy of diaphragm myofibers, and more studies using ultrasonography have shown that diaphragm atrophy and dysfunction are frequent complications that occur during mechanical ventilation and are associated with poor outcomes.”  The “and” in the middle needs to be removed to make this sentence align with standard English grammar as follows: “Since Levine et al. reported in 2008 that short duration-controlled ventilation in brain-dead organ donors causes marked atrophy of diaphragm myofibers, more studies using ultrasonography have shown that diaphragm atrophy and dysfunction are frequent complications that occur during mechanical ventilation and are associated with poor outcomes.”

[Response]

We removed the “and” as you suggested.

Lines 66-67: “Importantly, in critically ill patients with systemic inflammation, sarcolemma is 66 vulnerable to mechanical stimuli, therefore diaphragm dysfunction through this mecha-67 nism frequently happens20.”  This sentence is oddly worded and could be rephrased as something like this: “Importantly, in critically ill patients with systemic inflammation, mechanical stimuli can exacerbate sarcolemma and thus contribute to diaphragmatic dysfunction.”

[Response]

We rephrased the sentence as suggested.

Reviewer 2 Report

Comments and Suggestions for Authors

I have the following suggestions about this paper:

1.       The author may discuss the particular drawbacks (invasiveness, affordability, and ease of use at the bedside) for diagnostic ultrasonographic devices surpass in comparison to alternative technologies and provide evidence to support the statement that they are the best tools for observing and measuring diaphragm morphology and function.

2.       The author should explain the mechanisms that account for the diaphragm's selective atrophy and mitochondrial dysfunction in brain-dead patients as opposed to the pectoralis major and discuss about the evidence to the involvement of proteolytic pathways in this process.

3.       The author may illustrate the current clinical or experimental research about how diaphragmatic function is affected by eccentric contraction during expiratory muscle recruitment, especially in patients who have small airway obstruction.

4.       The author may provide examples about what other techniques or methods can be applied to accurately assess the work of breathing and diaphragm function in patients on mechanical ventilation for the limitations of excursion measurement during unassisted spontaneous breathing.

5.       The author should discuss about the methods can be used to increase the left diaphragm's visibility during ultrasonography, and how these methods can be used to improve the precision of the diagnosis of diaphragm dysfunction in patients.

Author Response

Reviewer 2

Comments and Suggestions for Authors

I have the following suggestions about this paper:

  1. The author may discuss the particular drawbacks (invasiveness, affordability, and ease of use at the bedside) for diagnostic ultrasonographic devices surpass in comparison to alternative technologies and provide evidence to support the statement that they are the best tools for observing and measuring diaphragm morphology and function.

[Response]

In the original manuscript, we did not claim that ultrasound is the best tool in assessing diaphragm function and described its limitations in detail. Also, we cited expert consensus to objectively assess the potential value of diaphragm ultrasonography.

  1. The author should explain the mechanisms that account for the diaphragm's selective atrophy and mitochondrial dysfunction in brain-dead patients as opposed to the pectoralis major and discuss about the evidence to the involvement of proteolytic pathways in this process.

[Response]

This manuscript is not a review of ventilator-induced diaphragm injury (VIDD) but diaphragm ultrasound. We believe that our concise summaries of each mechanism of VIDD with references is working in this current form.

  1. The author may illustrate the current clinical or experimental research about how diaphragmatic function is affected by eccentric contraction during expiratory muscle recruitment, especially in patients who have small airway obstruction.

[Response]

Same as above.

  1. The author may provide examples about what other techniques or methods can be applied to accurately assess the work of breathing and diaphragm function in patients on mechanical ventilation for the limitations of excursion measurement during unassisted spontaneous breathing.

[Response]

We added the statement “…, and TFdi can work for that purpose”. The ability of TFdi to estimate respiratory effort has been discussed in “3. Evaluation of respiratory effort” in the section of Clinical applications of diaphragm ultrasonography.

  1. The author should discuss about the methods can be used to increase the left diaphragm's visibility during ultrasonography, and how these methods can be used to improve the precision of the diagnosis of diaphragm dysfunction in patients.

[Response]

Thank you for your suggestion. According to the comment from reviewer 1, we added description about novel approach being able to visualize the left diaphragm reliably.

Round 2

Reviewer 1 Report

Comments and Suggestions for Authors

Overall, the Authors have acknowledged the approach I have identified was missing from their paper (Da Conceicao D, et al).  The absence of this approach from a consensus statement on diaphragmatic ultrasound is more a failure of the consensus statement than of the approach.  However, the new paper is an adequate improvement over the first draft, such that I do not object to publication.